# Influence of Surface Roughness on the Properties of Nitrided Layer on 42CrMo4 Steel

**DOI:** 10.3390/ma16134496

**Published:** 2023-06-21

**Authors:** Marcin Moneta, Jerzy Stodolny, Beata Michalkiewicz, Rafał Jan Wróbel

**Affiliations:** 1Department of Chemical Technology and Engineering, West Pomeranian University of Technology, Szczecin, ul. Pułaskiego 10, 70-322 Szczecin, Poland; beata.michalkiewicz@zut.edu.pl (B.M.); rafal.wrobel@zut.edu.pl (R.J.W.); 2Elterma S.A. S.K.A., ul. Poznańska 58, 66-200 Świebodzin, Poland; 3Independent Researcher, ul. Pisza 27, 32-700 Bochnia, Poland; jerzy.stodolny@gmail.com

**Keywords:** nitriding, roughness, surface, materials engineering, thermochemical treatment

## Abstract

A crucial factor of a nitriding process of treated parts is surface roughness. Eight samples of 42CrMo4 steel were used to investigate the parameter represented by Ra. In the study, the innovative combined microhardness profiles were used to present results within the compound zone and diffusion layer. Therefore, two loads were applied in the compound zone, 5 gf, and diffusion layer, 500 gf. Observation with SEM and chemical analysis of the investigated samples showed a correlation between microstructure, nitrogen concentration and microhardness of the compound zone. XRD diffraction was used to identify the phase composition. Moreover, the X-ray photoelectron spectroscopy technique was also applied in the study. No distinct correlations between compound zone morphology and the Ra parameter were observed. The thickness value of the structure was constant and fluctuated around 20 µm in the vast majority of the examined cases. However, analysis of the results revealed a dependence between the Ra parameter and diffusion layer thickness. The values of this parameter varied in the range of 356–394 µm depending on the Ra parameter. A distinct nitrided layer was observed on the polished sample.

## 1. Introduction

One of many advantages of nitriding is increasing corrosion resistance, which is attributed to the external compound zone [1,2]. Another important property of nitrided layers is the improvement of fatigue strength resulting from the diffusion layer [1,3]. Nitriding compared to carburizing is characterized by lower processing temperature and significantly smaller distortions [4].

A huge number of factors determine a nitriding process. Parameters such as temperature [5], nitriding potential [6], time [7] or pressure value [8] influence the complex thermochemical treatment. Because a nitriding process is associated with catalytic decomposition of ammonia, another significant factor is the surface condition of a treated part.

All residues from previous treatments should be eliminated before a nitriding process. Particles of cooling thermal grease, detergent and oxide layers can decrease absorption on a nitrided surface. It is also recommended to avoid residues of nonmetallic parts such as lead, tin, zinc or copper [1].

Surface roughness can be represented by many different parameters, one of which is the arithmetic average height—Ra. This parameter is easy to define, measure and provides a good indication of the variation in profile height, making it a universal indicator in overall quality control. Surface roughness can also be represented by other parameters. One of them is Rz, which represents the average values of the five deepest points and five highest peaks within an elementary section of the surface profile. Another possible indicator is Rm, which represents the maximum height of the profile, measured as the distance between its highest and lowest points within a specified measurement interval [9,10].

Surface condition has an important effect on catalytic decomposition of NH_3_. It is widely claimed that a polished surface inhibits ammonia decomposition and increases the initial temperature value of the process up to 600 °C [11]. A surface activation is recommended before nitriding treatment. Mechanical methods are used for surface activation. In the group, we can distinguish techniques such as shot peeling, rolling, sandblasting or shot blasting. All of them increase the efficiency of diffusion saturation. Chemical reactions between the treated surface and constituents from the working atmosphere are facilitated due to energy from previous deformations [12]. Abrasive blasting also enables the removal of paint or solder residues and some imperfection of the surface. An increase in the surface roughness value is commonly observed after a nitriding process [13,14]. It is also claimed that a higher value of the parameter before nitriding causes higher growth during the process [1]. However, this is a complex phenomenon and depends on numerous factors. One of them is the kind of nitriding method used [15,16]. Nitriding gases usually cause higher modification of surface roughness than the plasma version of the thermochemical process. Based on the literature data, the main reason for the surface roughness modification is the appearance of a compound zone after nitriding. Negligible modification of the surface roughness should be observed after a nitriding process without the compound zone existing. A low decrease in the surface roughness value can also occur after a nitriding process [16,17,18]. A significant factor is the grade of the applied material [15,16]. Dobrocky et al. [19] investigated changes of surface roughness of 42CrMo4 steel after nitriding in plasma and gas. The results showed no changes in the Ra parameter after the nitriding process and small decrease in other roughness parameters such as Rz. Surface morphology influences corrosion resistance and wear resistance after an oxidation process. Polished and more regular surfaces increase the parameter values [20]. 

The steel 42CrMo4 is widely used in the machinery, aerospace and military industries. The material is applied in components such as gear wheels, crankshafts and barrel and breechblock mechanisms due to the high strength and high toughness properties of the steel grade. However, the steel has characteristics of unfavorable tribological properties due to its low abrasive resistance [19,21,22]. Surface properties of 42CrMo4 steel can be improved by thermochemical treatment—nitriding [23,24,25].

## 2. Materials and Methods

### 2.1. Materials

42CrMo4 steel was selected for the present experiment due to its popularity in nitriding processes [7,19,20,21,26,27]. Its chemical composition is shown in Table 1. The material was used in the delivery state. The previous heat treatment process contained hardening and high-temperature tempering at a 600 °C temperature value. Investigated samples were cube-shaped with a side length of 1cm. Samples were ground with different abrasive papers (P120, P180, P320, P600, P800, P1000, P2500). The grinding process was conducted applying a 300 rpm rotation speed value. Duration of the mechanical treatment was 30 s for each surface. One of the investigated samples was polished with the application of different diamond suspensions—9 µm, 6 µm and 3 µm.

The nitriding process was conducted in a furnace with a 500 mm length and a 32 mm inner diameter quartz pipe. The ammonia flow was 50 cm^3^/min. The steps of the nitriding process were as follows:Heating at the rate of 5 °C/min up to a temperature value of 540 °C in nitrogen atmosphere.Nitriding at 540 °C in the atmosphere of 5.0 pure ammonia and flow at 50 cm^3^/min. The duration of the process was 12 h.Cooling up to room temperature.

After the thermochemical treatment, the samples were prepared using chemical nickeling, cutting, hot embedding, grinding and polishing.

### 2.2. Characterization Techniques

Surface roughness has been measured with an optic 3D measuring instrument InfiniteFocusSL Alicona. Microhardness measurements were performed on a Vickers tester UHL VMHT AUTO using a load of different values within the compound zone and diffusion layer. Chemical composition was examined with the microanalysis X-ray method on an EDS detector (EDS NSS 312, ThermoScientific, Waltham, MA, USA), which was connected to a scanning electron microscope HitachiSu8020. Microstructure analyses were conducted on scanning electron microscopes—JEOL-6400 (JEOL, Tokyo, Japan) and HitachiSu8020 (Hitachi, Tokyo, Japan). X-ray diffraction (XRD) was conducted using an Empyrean (PANalytical, Malvern, UK) diffractometer with a monochromator and PIXcel3D detector. A Cu anode was used to obtain Cu K_α_ (K_α1_ = 0.154056 nm). The measurements were conducted in 2θ ranges of 20–80°. The PDF-4+ 2020 International Centre for Diffraction Data database and High Score Plus software were used for phase analysis. 

Chemical composition was determined with X-ray photoelectron spectroscopy in a commercial UHV surface analysis system (PREVAC, Rogow, Poland). The analysis chamber was equipped with nonmonochromatic X-ray photoelectron spectroscope and a kinetic electron energy analyzer (SES 2002; Scienta, Uppsala, Sweden). The XPS analysis was performed using Al K_α_ (h = 1489.6 eV) radiation.

## 3. Results

### 3.1. Surface Roughness

3 Ra measurements were conducted for each variant. The analysis was based on linear profiles. Exemplary images of the surfaces and their roughness profiles are shown in Figure 1. All obtained roughness values are in Table 2 and Figure 2.

### 3.2. Microhardness Measurements

In the article, an alternative presentation of microhardness profiles was shown. Usually, the measurements are carried out only within the diffusion layer. The approach limits information of a compound zone structure and its porosity or phase composition on the mechanical property. Moreover, the proposed solution enables researchers to obtain more accurate microhardness measurements of the described layer due to limitations of the impact of the substrate material. Microhardness measurements were performed in the oblique path relative to the surface and with a space of 2 μm within the compound zone. The solution is shown Figure 3. The measurements were conducted using a 5 gf load.

The microhardness measurements were performed using a 500 gf load and with a space of 50 μm within the compound zone. The described method is illustrated in Figure 4. The approach enabled researchers to obtain combined microhardness profile graphs for the entire nitrided layer and estimate its thickness according to DIN 50190-3.

Combined graphs for each variant are presented in Figure 5.

Three optical measurements of each investigated compound zone thickness and their respective porous parts were conducted. The results can be observed in Table 3 and Figure 6.

### 3.3. Analysis with Scanning Electron Microscope

During the investigation procedure, observations with a scanning electron microscope were conducted. Due to similarity of the obtained morphology of the investigated samples, only one case of the structure was presented. Figure 7 shows the exemplary compound zone forming during the nitriding process. 

### 3.4. Chemical Composition

Investigation of nitrogen concentration was conducted using linear analysis. Figure 8 shows the result combined with its microhardness profile and microstructure image. The applied approach shows dependencies between all of the parameters within the nitrided layer. 

### 3.5. Phase Analysis with XRD Diffractometer

Figure 9 shows an exemplary XRD pattern. Due to the limitation of X-ray penetration inside the material, only the external surface of the compound zone was investigated. 

### 3.6. XPS

The results of XPS analysis can be observed in Figure 10. The signal obtained after nitriding was divided into two peaks. The first of them was registered with the binding energy value of 400.3 eV and the second with the energy of 397.6 eV. 

## 4. Discussion

Abrasive paper with various gradation enabled researchers to obtain different surface roughness values represented quantitatively by the Ra parameter. Applying a higher value of the abrasive paper decreased the parameter, which is shown in Table 2 and Figure 2. A decrease in the Ra parameter was observed in each investigated case. The described phenomenon could be caused by a thin iron oxide film on the nitrided surface. According to the article of Baranowska [28], oxygen can have an influence on decreases in the Ra parameter. 

Analysis of microhardness profiles within compound zones presented a repeat Table 3 stages tendency shown in the Figure 5. In the first, microhardness value increases due to crossing from the porous zone to more homogenous structure. Subsequently, the parameter obtained its maximum value in the range of 900–1000 HV0.005. In the last stage, the microhardness value decreased due to progressive crossing to the diffusion layer. Unequivocal dependence between surface roughness and compound zone thickness was not observed—Table 3 and Figure 5. However, the compound zone thickness of the polished state was significantly lower than in case of the other variants. The alternative presentation of microhardness in the form of combined graphs has shown a broader sense of the nitrided layer. The described approach presented the detailed influence of porosity on the mechanical value. The repeatability of the curves proved its reliability. The method should be more beneficial for the compound layers with different structures and properties. A porous zone was observed in each case and constituted between 40 and 50% of the entire compound zone thickness. Compound zone thickness values and their porous parts were characterized by constant level of 18.1–21.0 μm and 8.5–9.6 μm, respectively. In the Figure 7, a slight oxide layer can be observed, which was formed during the cooling process.

Observation of microhardness profiles within the diffusion layer revealed the decreasing of the parameter up to the core microhardness value of 300 HV0.5—Figure 5. Analysis of the data enables researchers to perceive a distinct dependence between surface roughness and diffusion layer thickness, shown in Table 3 and Figure 6. An increasing in Ra caused the simultaneous growth of diffusion layer thickness, which was the most distinct up to approximately a 0.33 μm value of Ra. After the level, the growth was emphatically lower. The phenomenon is caused by an increase in the extended surface of the investigated samples and extension of the boundary NH_3_/surface [7]. The thickness of the diffusion layer (NHD) can be described according to the following equation:(1)$$NHD=k\sqrt{t},$$
where *k* is a temperature-dependent constant and *t* is time of nitriding [29,30].

Based on the relationship above, the growing rate of the diffusion layer decreases with the time of the nitriding process. The obtained differences could be more distinct in the case of applying a shorter duration of nitriding.

The nitrogen profile combined with previous investigation in Figure 8 enabled researchers to present the correlation between microstructure, nitrogen concentration and microhardness within the compound zone. The layer is rich in nitrogen and distinctly separate from the diffusion one. Additionally, the compound zone can be divided into two layers based on the nitrogen content. The external part is richer in nitrogen, whereas the internal one has a lower content of the element. The obtained differences indicate the dual-phase content of the compound zones, which consist of external ε(Fe_2_N_(1+x)_) and internal γ′ (Fe_4_N) [23]. The microhardness path revealed a significant correlation in relation to the microstructure. The initial low microhardness value increased due to passing through the porous zone to homogenous space. The subsequent step was stabilization of the microhardness value to approximately 900 HV0.005. Finally, the parameter decreased to 550 HV0.005 due to crossing the diffusion layer with a lower concentration of nitrogen.

The XRD pattern in Figure 9 revealed the existence of the epsilon phase in the external part of the compound zone. Analysis of phases in the deeper part of the layer could not be conducted due to significant X-ray absorption by the sample up to a depth of approximately 5 μm. The obtained results proved the dual-phase composition of the investigated compound zones. In the range of the structure zone, the investigated compound zone consists of an external phase ε and γ′ in its internal part.

The XPS analysis in Figure 10 revealed the signal N 1 s. In comparison with the non-nitrided case, a significant signal from nitrogen can be observed for the nitrided sample. The signal consists of two components. The first is from a nitrogen and oxygen compound with the component position at about 400.5 eV. The second component can be observed at about 397.5 eV from iron nitride. It is worth noting that the XPS method is very surface-sensitive and delivers information from the depth of about 1.0 nm. Therefore, the oxygen contribution may be the result of passivation of the surface after the nitriding process.

The investigations were conducted on 42CrMo4 steel. The chemical composition of the processed material can influence the relationship between the Ra parameter and the parameters of the nitrided layer. The described investigations can be continued to define the impact of the chemical composition of steels and non-ferrous material on the phenomena under consideration.

## 5. Conclusions

Surface roughness decreased its value after the nitriding process in the case of each variant. Existence of a slight internal oxide layer could have influenced the phenomenon.

Correlation between surface roughness and compound zone thickness was not observed. The maximum value for the layer was approximately 900–1000 HV0.005.

However, a distinct correlation between diffusion layer thickness and the Ra value was noted. An increasing in the parameter causes asymptotic growth of the diffusion layer. The significant growth can be observed up to a 0.33 μm Ra value. The subsequent increasing of the parameter causes only a slight growth of diffusion layer thickness. A distinct nitrided layer was observed in the case of the polished sample, which is in opposition to the literature data [6].

Innovative microhardness combined graphs were determined within the compound zone and diffusion layer. The approach enables the characterization of the entire nitrided layer instead of the only the diffusion part. In the case of the compound zone, a similar pattern of the microstructure graph was observed. No correlation between the compound zone and surface roughness was observed even in case of the detailed investigation.

The XRD pattern indicates a dual-phase compound zone consisting of the external ε phase and the internal γ’ one.

## Figures and Tables

**Figure 1 materials-16-04496-f001:**
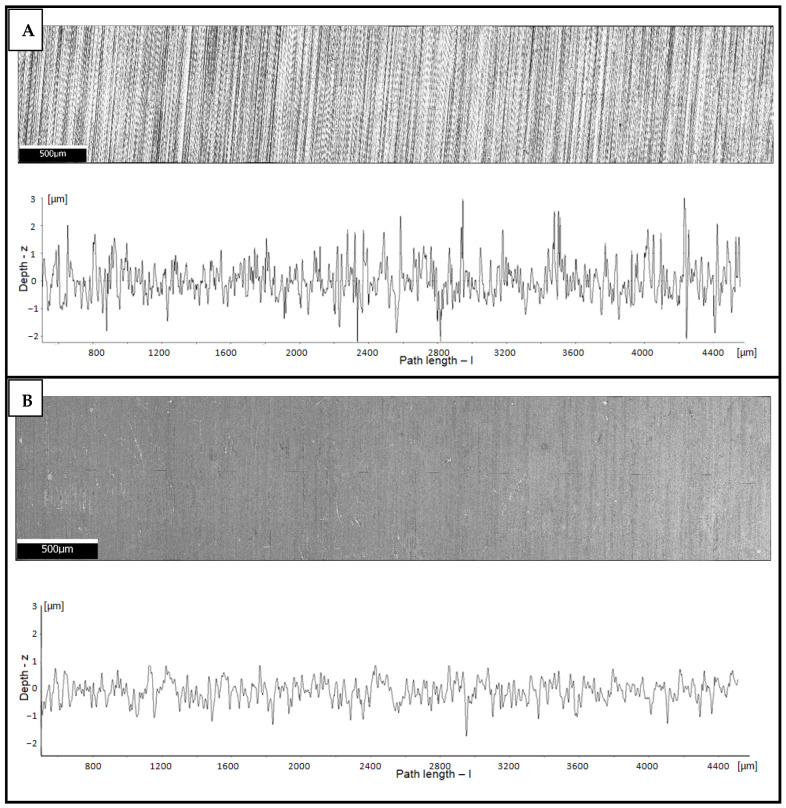
Exemplary images of surfaces and their roughness profiles for the P320 ground sample: before (**A**) and after nitriding (**B**).

**Figure 2 materials-16-04496-f002:**
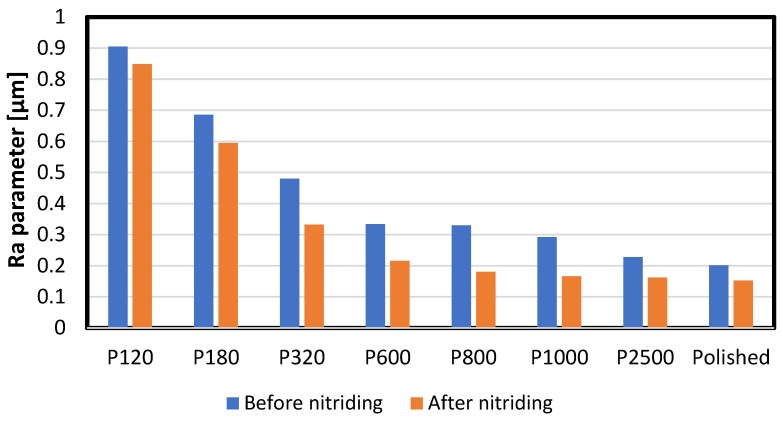
Average Ra parameter value for each case.

**Figure 3 materials-16-04496-f003:**
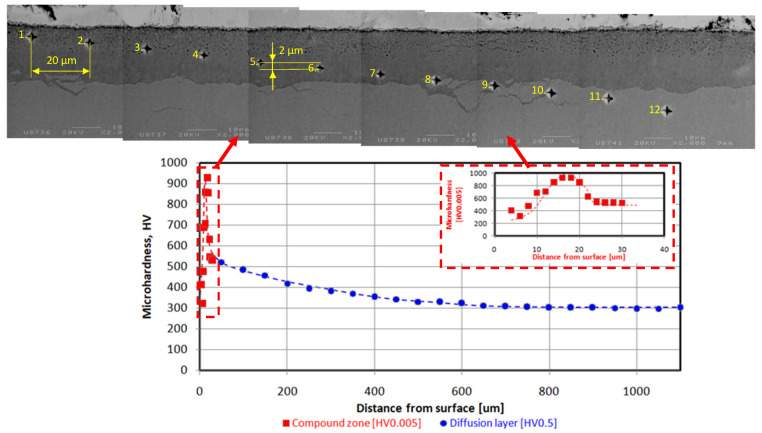
Microhardness profile within the compound zone and its image in the combined graph. The numbers 1–12 describe the order of performing measurements.

**Figure 4 materials-16-04496-f004:**
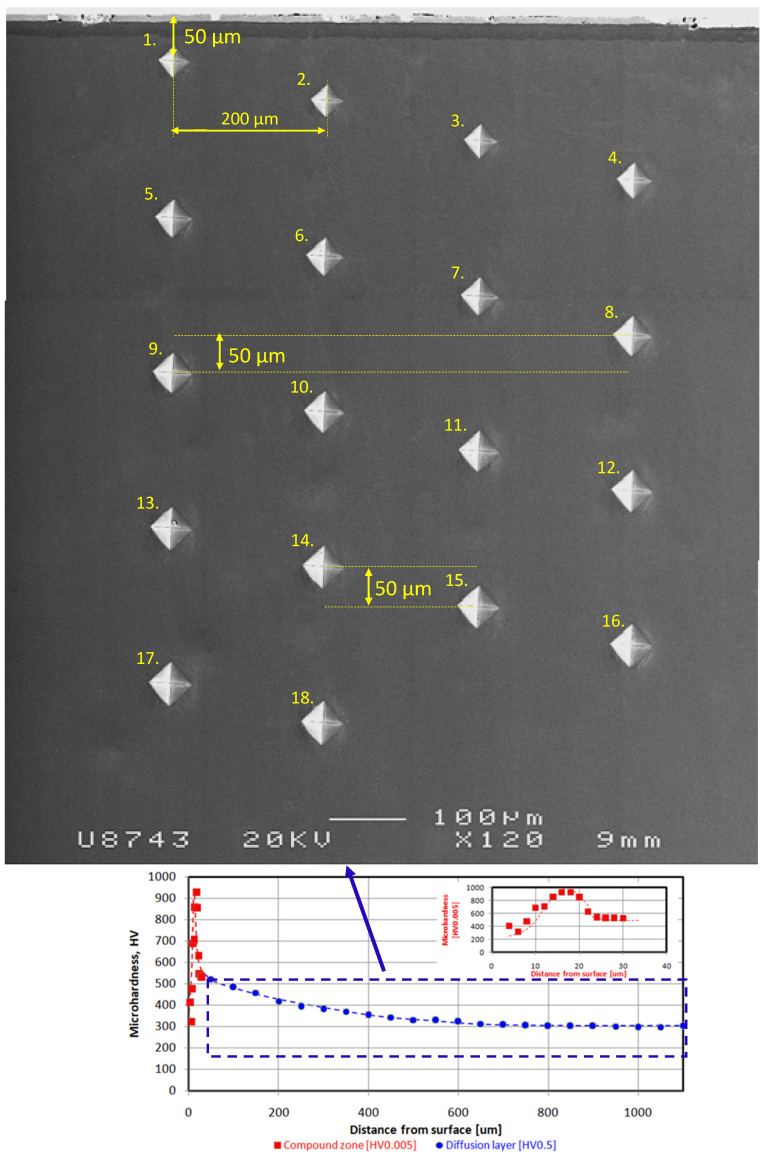
Microhardness profile within diffusion layer and its image in the combined graph. The numbers 1–18 describe the order of performing measurements.

**Figure 5 materials-16-04496-f005:**
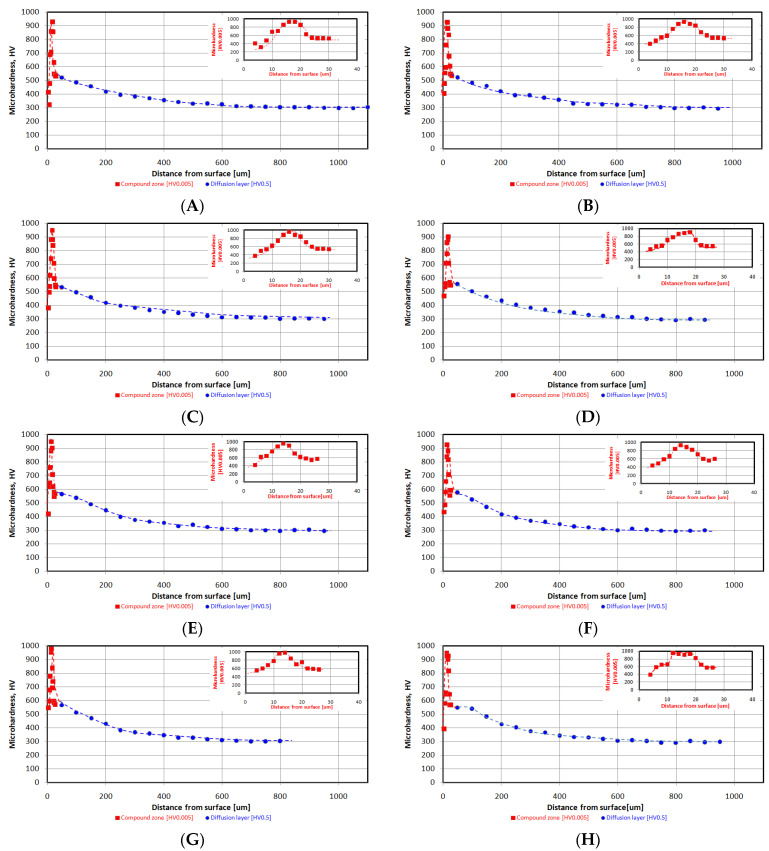
Microhardness combined graphs within the compound zone and diffusion zone for the variants as follows: (**A**)—P120, (**B**)—P180, (**C**)—P320, (**D**)—P600, (**E**)—P800, (**F**)—P1000, (**G**)—P2500, (**H**)—polished.

**Figure 7 materials-16-04496-f007:**
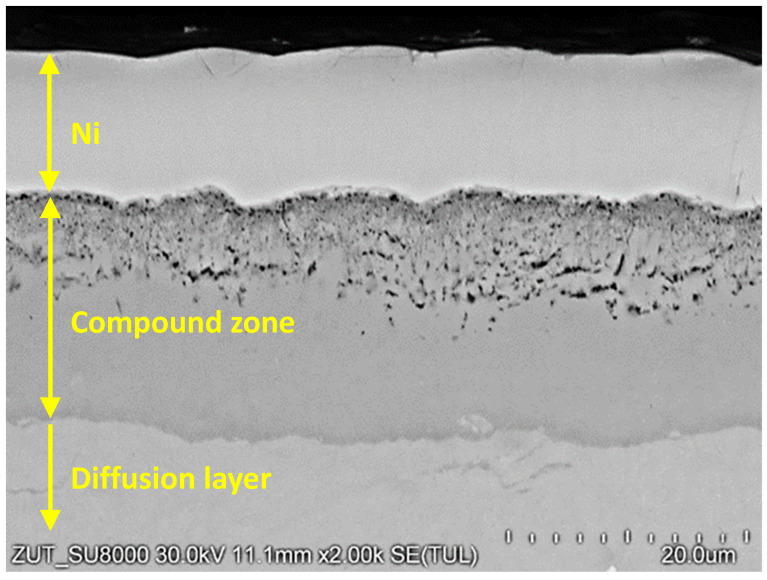
Exemplary SEM image of compound zone conducted on microscope SEM HitachiSu8020.

**Figure 8 materials-16-04496-f008:**
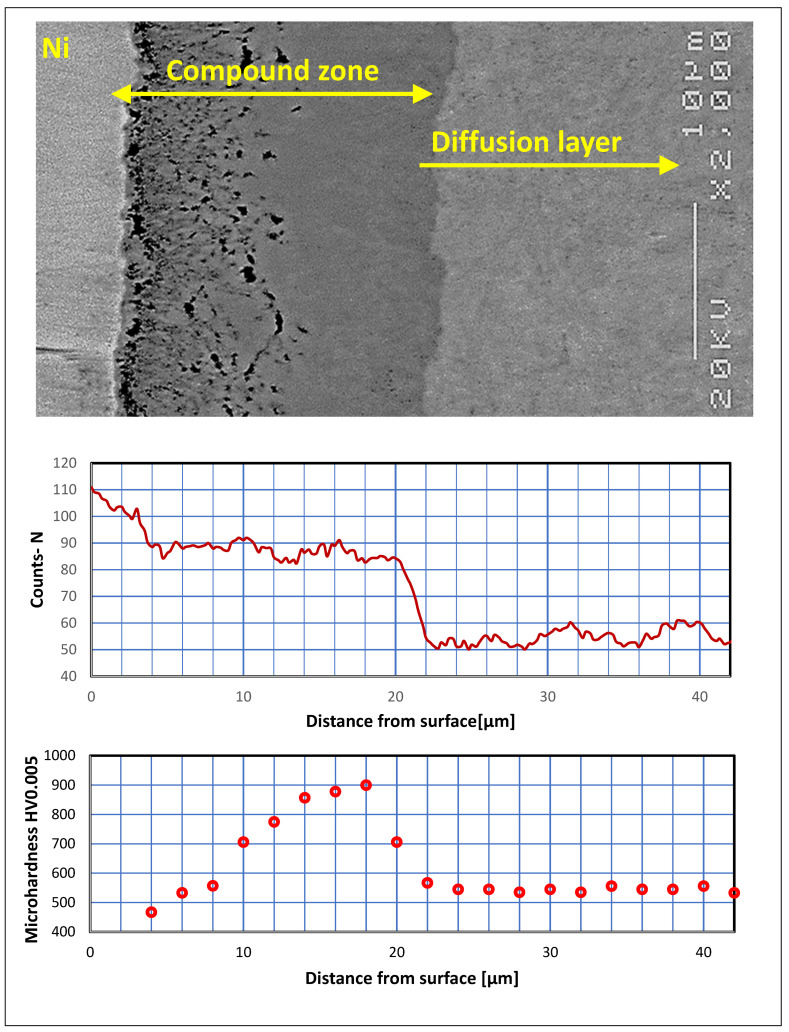
SEM image of compound zone structure (SEM JEOL-6400) combined with its nitrogen and microhardness profiles.

**Figure 9 materials-16-04496-f009:**
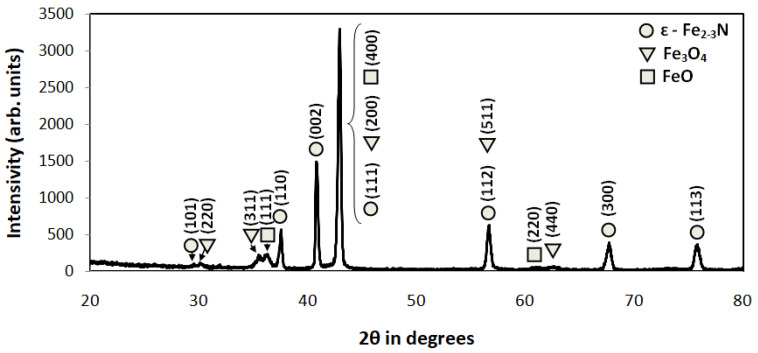
XRD pattern of the sample ground with P600 abrasive paper.

**Figure 10 materials-16-04496-f010:**
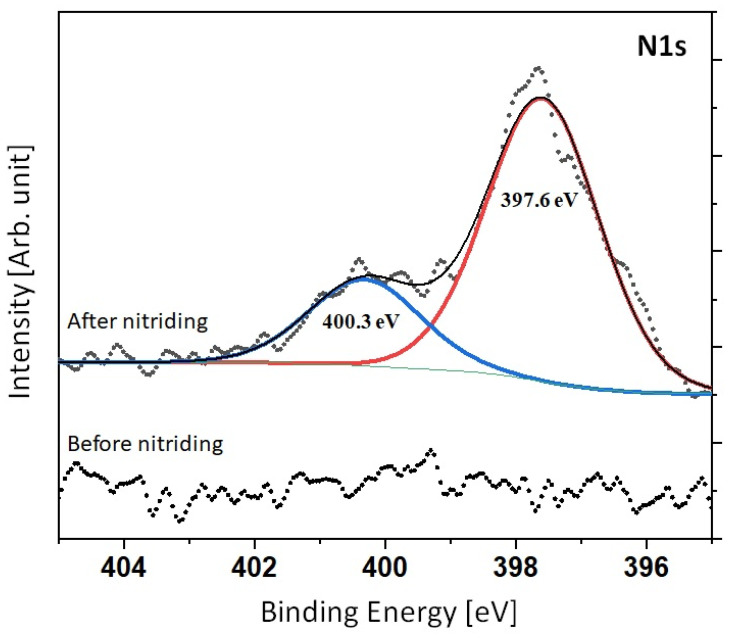
XPS pattern of the investigated surface.

**Figure 6 materials-16-04496-f006:**
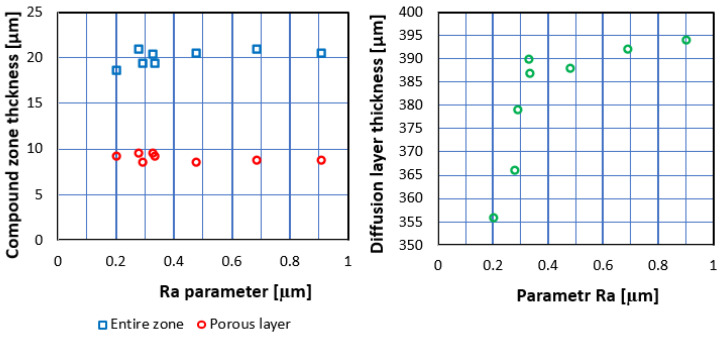
Effect of surface roughness (Ra) on compound zone thickness and diffusion layer thickness.

**Table 1 materials-16-04496-t001:** Chemical composition (wt.%) of 42CrMo4 steel.

C	Cr	Mn	Si	Mo	Ni	V	W	S	P	Fe
0.38–0.45	0.9–1.2	0.4–0.7	0.17–0.37	0.15–0.25	max 0.3	max 0.05	max 0.2	max 0.035	max 0.035	base

**Table 2 materials-16-04496-t002:** Average Ra parameter value for each case.

Surface Condition	Average Ra Parameter Value [µm]	
	Before Nitriding	After Nitriding
Abrasive paper P120	0.905	0.849
Abrasive paper P180	0.686	0.595
Abrasive paper P320	0.480	0.332
Abrasive paper P600	0.334	0.216
Abrasive paper P800	0.330	0.180
Abrasive paper P1000	0.292	0.166
Abrasive paper P2500	0.282	0.162
Polished	0.201	0.152

**Table 3 materials-16-04496-t003:** Thickness of the investigated nitrided layers.

Surface Condition	Average Thickness Value of Porous Zone [μm]	Average Thickness Value of Compound Zone [μm]	Thickness Value of Diffusion Layer [μm]
P120	8.7 ± 0.2	20.6 ± 1.3	394
P180	8.7 ± 1.2	21.0 ±1.9	392
P320	8.5 ± 0.7	20.5 ± 1.3	388
P600	9.2 ± 0.4	19.5 ± 0.3	387
P800	9.6 ± 0.3	20.3 ± 0.4	390
P1000	8.4 ± 0.8	19.5 ± 1.4	379
P2500	9.6 ± 1.3	21.0 ± 0.6	366
Polished	9.2 ± 0.3	18.6 ± 0.6	356

## Data Availability

Not applicable.

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
