# Peer review of "Influence of Surface Roughness on the Properties of Nitrided Layer on 42CrMo4 Steel"

_materials, 2023, doi:10.3390/ma16134496_

Round 1
Reviewer 1 Report
materials-2411210-peer-review-v1
Recommendation: revision
The authors are interested to study the Influence of surface roughness on the properties of nitrided layer on 42CrMo4 steel. The idea is interesting, but the following points need to take care in order to establish the usefulness of the article.
What could be the industrial application of the finding of the manuscript?
How Ra is influenced by the composition of the steel or any other non-ferrous alloy?
The authors should carry out similar experiments for 8-10 different composition of steel and non-ferrous alloy to deduct any co-relation between the composition of the subject and the Ra values.
Also, the sentence structure is not good, such as in the abstract authors are writing: A crucial factor of nitriding is the surface roughness of the treated parts. Nitriding of what?
However, the analysis of the results revealed a dependence between Ra parameter and the diffusion layer. What properties they are talking about here?
need to improve
Reviewer 2 Report
Dear Authors,
As a result of my review, I have the following comments and recommendations about the Manuscript ID: materials-2411210 completed by Marcin Moneta, Jerzy Stodolny, Beata Michalkiewicz and RafaÅ‚ Jan Wróbel refers to the “Influence of surface roughness on the properties of nitrided layer on 42CrMo4 steel”. Many methods have been used in order to improve significantly fatigue strength and corrosion resistance of ferrous metals. Among them, thermochemical process or nitriding was investigated in this study.
The scientific manuscript presents a compelling study that investigates the influence of surface roughness on the properties of nitrided layers. The chosen material for this investigation is 42CrMo4 steel, a widely used steel grade for nitriding purposes and a suitable representative material for the study. The authors emphasize the importance of surface roughness as a key parameter that affects both the tribological properties and the nitriding process.
The experimental work conducted by the authors is comprehensive and involves variations in Ra (roughness average) during the nitriding process. Various characterization methods, including advanced techniques such as X-ray diffraction (XRD) and X-ray photoelectron spectroscopy (XPS), were employed to obtain accurate data. The presented data effectively demonstrate the relationships between surface roughness and the properties of nitrided layers. An innovative approach to presenting micro-hardness profiles is introduced in the article, enabling the characterization of the mechanical properties of both the diffusion layer and the compound zone. Although the potential of this approach was not fully realized in the investigated case, it serves as a promising method for future studies.
However, there are a few areas and remarks within the reviewed work that could be improved.
1) Certain content, such as Figure 5, could be omitted without significantly affecting the overall article.
2) Additionally, further elaboration on the implications of the presented results and stronger conclusions derived from detailed considerations would enhance the impact of the article within the field.
3) Abstract: some important results (data) should be added just to get a clear idea about the whole work.
4) Abreviations such as Rz, Rsk, Rku should be defined in introduction section
5) Among other types of steel, Why 42CrMo4 has been investigated in this study ?
6) The advantages of the nitriding method should be highlighted in comparison to the other coated methods, which would enhance the interest in study of such a kind of steel.
7) A table should be included to compare you results with other materials used for the same purpose
In conclusion, this manuscript provides a valuable contribution to the research field of the nitriding process. The authors conducted well-designed experimental investigations and effectively presented their results in a clear and concise manner. Therefore, I recommend the publication of this manuscript in MDPI Materials, provided that the aforementioned remarks are adequately addressed.
Author Response
Dear Reviewer,
Thank you so much for taking the time to write the review. We appreciate your feedback and are grateful for your support.
We want to express our gratitude for the constructive criticism and suggestions you shared in your review. Your insights are incredibly valuable to us as they help us identify areas where we can improve and enhance our article.
In response to your suggestions, we have thoroughly revised the article you reviewed and have attached the revised version to this message. In the revised article, we have carefully addressed the areas you highlighted and made the necessary changes to ensure that the content meets your expectations. To make it easier for you to identify the modifications, we have clearly marked the revised sections within the document.
Q -Certain content, such as Figure 5, could be omitted without significantly affecting the overall article.
A - Based on your recommendations, we reviewed the article thoroughly and removed some insignificant content
Q- Additionally, further elaboration on the implications of the presented results and stronger conclusions derived from detailed considerations would enhance the impact of the article within the field.
A - After additional analysis of the discussion of the results and the received reviews, we have identified a lack of information regarding the potential influence of the chemical composition on the obtained results. This fact has been added in the discussion section.
Q- Among other types of steel, Why 42CrMo4 has been investigated in this study ?
42CrMo4 steel is a popular grade for nitriding. We have added additional literature references to emphasize this fact.
Q - Abreviations such as Rz, Rsk, Rku should be defined in introduction section
A - We have defined the most popular surface roughness parameters in the introduction section.
Q - Abstract: some important results (data) should be added just to get a clear idea about the whole work.
A - Following the recommendation, we have added information about significant results in the abstract.
Q - The advantages of the nitriding method should be highlighted in comparison to the other coated methods, which would enhance the interest in study of such a kind of steel.
A - We presented the main advantages of nitriding and compared it to another popular type of treatment - carburizing.
Q - A table should be included to compare you results with other materials used for the same purpose
A - The chemical composition of the processed material can influence the relationship between the Ra parameter and the parameters of the nitrided layer. However, nitriding is a complex process that depends on multiple factors. Such a comparison should include processes in which the only difference would be the chemical composition of the materials. Other process parameters such as temperature, ammonia flow rate, atmosphere composition, and process duration would remain the same. The potential impact of composition on the obtained results was highlighted in the discussion section. The mentioned remark represents an extremely interesting issue that deserves further investigation in subsequent publications.
Once again, thank you for your review. We genuinely value your support and are grateful to have reviewer like you who help us improve. If you have any further suggestions, please feel free to reach out to us.
Best regards,
Marcin Moneta,
RafaÅ‚ Wróbel,
Jerzy Stodolny,
Beata Michalkiewicz
Round 2
Reviewer 1 Report
The manuscript could be accepted